# Thrombophilia-Related Single Nucleotide Variants and Altered Coagulation Parameters in a Cohort of Mexican Women with Recurrent Pregnancy Loss

**DOI:** 10.3390/diagnostics15243111

**Published:** 2025-12-07

**Authors:** Luis Felipe León-Madero, Larissa López-Rodriguez, Mónica Aguinaga-Ríos, Samuel Vargas-Trujillo, Angélica Castañeda-de-la-Fuente, Paloma del Carmen Salazar-Villanueva, Yanen Zaneli Ríos-Lozano, Yuridia Martínez-Meza, Monserrat Aglae Luna-Flores, Alberto Hidalgo-Bravo, Héctor Jesús Borboa-Olivares, Verónica Zaga-Clavellina, Rosalba Sevilla-Montoya

**Affiliations:** 1Department of Genetics and Human Genomics, National Institute of Perinatology, Montes Urales 800, Lomas-Virreyes, Lomas de Chapultepec IV Secc, Miguel Hidalgo, Mexico City 11000, Mexico; luisfe_leon@comunidad.unam.mx (L.F.L.-M.); dra.larissalopezr@gmail.com (L.L.-R.); aguinagamonica09@gmail.com (M.A.-R.); angiecast.95@gmail.com (A.C.-d.-l.-F.); paloma060290@gmail.com (P.d.C.S.-V.); yanenrios@gmail.com (Y.Z.R.-L.); yurimtz@live.com.mx (Y.M.-M.); aglae.lunaflores@gmail.com (M.A.L.-F.); 2Department of Critical Care Medicine, National Institute of Perinatology, Montes Urales 800, Lomas-Virreyes, Lomas de Chapultepec IV Secc, Miguel Hidalgo, Mexico City 11000, Mexico; samuelvargast@hotmail.com; 3Research Management Office, National Institute of Rehabilitation (INRLGII), Calzada Mexico-Xochimilco 289, Arenal de Guadalupe, Mexico City 14389, Mexico; dr_genetica@yahoo.com; 4Research Branch, National Institute of Perinatology, Montes Urales 800, Lomas-Virreyes, Lomas de Chapultepec IV Secc, Miguel Hidalgo, Mexico City 11000, Mexico; hector.borboa@inper.gob.mx; 5Research Management Office, National Institute of Perinatology, Montes Urales 800, Lomas-Virreyes, Lomas de Chapultepec IV Secc, Miguel Hidalgo, Mexico City 11000, Mexico; v.zagaclavellina@gmail.com

**Keywords:** recurrent pregnancy loss, thrombophilia, F12, F7

## Abstract

**Background:** Recurrent pregnancy loss (RPL) is a multifactorial condition in which genetic variants associated with thrombophilia may contribute to altered coagulation and adverse pregnancy outcomes. **Objective:** This study aimed to investigate the association between thrombophilia-related single nucleotide variants (SNVs) and coagulation-related metabolites in a cohort of Mexican women with RPL. **Methods:** A retrospective and descriptive design was conducted including 105 women with at least two consecutive miscarriages and with a multidisciplinary approach that included a thrombophilia-associated SNVs panel. Peripheral blood samples were collected after fasting for biochemical and molecular analyses. Genotyping of thrombophilia-associated SNVs was performed using real-time PCR with custom-designed TaqMan probes on a Rotor-Gene Q platform, including variants in AGT (rs4762, rs699), F7 (rs6046), FGB (rs1800790), MTR (rs1805087), MTRR (rs1801394), MTHFR (rs1801133, rs1801131), F2 (rs1799963), F5 (rs6025), SERPINE1 (rs1799889), F12 (rs1801020), and F13A1 (rs5985) genes. Coagulation parameters evaluated were folic acid, cobalamin, fibrinogen, D-dimer, homocysteine, antithrombin III activity, thrombin time (TT), prothrombin time (PT), activated partial thromboplastin time (aPTT), international normalized ratio (INR), and Factor XII activity. **Results:** Significant differences were found in INR values across F7-rs6046 genotypes (*p =* 0.006), with an additive model showing a mean difference of 0.05 (*p =* 0.0009). The F12-rs1801020 variant was strongly associated with Factor XII activity (*p =* 0.002) and aPTT (*p =* 0.045). **Conclusions:** These findings indicate that F7-rs6046 and F12-rs1801020 genotypes influence specific coagulation parameters, suggesting that certain thrombophilia-associated SNVs may modulate the hemostatic profile in Mexican women with RPL and contribute to personalized risk assessment in reproductive medicine.

## 1. Introduction

Recurrent pregnancy loss (RPL), most often defined as two or more consecutive spontaneous miscarriages before 20–24 weeks of gestation, affects approximately 1–5% of women of reproductive age [1]. Despite exhaustive investigations for uterine anomalies, parental karyotypes, endocrinopathies, autoimmune conditions (e.g., antiphospholipid syndrome), and infectious or hormonal causes, up to 50–75% of RPL cases remain unexplained [1,2]. This high proportion of idiopathic cases has led researchers to scrutinize additional contributory pathways, among which inherited thrombophilia and coagulation dysregulation are compelling candidates.

The concept that hemostatic imbalances may predispose to early pregnancy failure rests on the notion that subclinical microthrombi, impaired uteroplacental perfusion, or dysregulated fibrin turnover might compromise trophoblast invasion or placental development. In this paradigm, common single nucleotide variants (SNVs) in genes of the coagulation cascade may subtly modulate levels or activity of clotting factors, thereby shifting the systemic or local balance toward a prothrombotic state. Integrating genotypes with functional coagulation phenotypes (e.g., clotting times, factor activity) offers a promising strategy to move beyond mere association toward mechanistic insight. Despite multiple studies addressing classical thrombophilia markers (e.g., Factor V Leiden, prothrombin G20210A, MTHFR 677C>T), their explanatory power in RPL cohorts has been modest and inconsistent [1,3,4]. Moreover, many studies lack concurrent measurement of coagulation phenotypes, limiting the ability to link genetic variation to functional consequences. Beyond these canonical variants, components of the contact pathway, such as coagulation Factor XII (FXII), have attracted renewed attention. Recent work highlights potential links between FXII activity and adverse pregnancy outcomes, suggesting that functional alterations in this pathway may contribute to pregnancy loss [3,4]. These insights align with a broader view that coagulation parameters (e.g., INR, aPTT) and related biochemical markers (e.g., fibrinogen, D-dimer, folate, homocysteine) can serve as phenotypic readouts of underlying genetic variation influencing hemostasis [2,5,6].

From this perspective, two single-nucleotide variants assumed relevance. On gene F7 the SNV rs6046 (Arg353Gln, also known as R353Q) has been extensively studied in cardiovascular and thrombosis research. The allele encoding the amino acid Gln is associated with lower plasma FVII levels and activity in numerous populations [7]. However, only recently has this variant been tested in the context of RPL. In an Iranian study of 144 women with RPL vs. 150 controls, Kavosh et al. [7] reported a significant difference in genotype frequencies, with an odds ratio indicating a protective effect for the allele encoding the Gln in several genetic models. Nevertheless, that work did not incorporate measurements of INR, PT, or FVII activity in carriers vs. non-carriers. The second relevant variant was the SNV rs1801020 in the gene F12. It has been mechanistically linked to reduced translation and lower Factor XII levels/activity, and genome-wide association studies have shown that rs1801020 contributes significantly to interindividual variation in aPTT and FXII activity [1]. Yet, its direct role in RPL remains underexplored. To the best of our knowledge, there are no studies testing the association of the SNV rs1801020 with RPL.

Thus, the literature contains a precedent for rs6046 being associated with RPL risk, but without functional coagulation correlation, and a solid mechanistic basis for rs1801020 influencing clotting phenotypes but not previously investigated in RPL in tandem with coagulation metrics.

Given that Mexican and Latin American populations remain underrepresented in thrombophilia/RPL research and considering potential allele frequency differences and gene-environment interactions there is a distinct rationale for focusing on these two SNVs in a Mexican RPL cohort with measurement of coagulation parameters [8]. Herein, by genotyping F7-rs6046 and F12-rs1801020 alongside a panel of other thrombophilia-related SNVs and correlating them with a complete coagulation profile (INR, PT, aPTT, TT, fibrinogen, D-dimer, homocysteine, antithrombin III, FXII activity), we aim to bridge the gap between genotype and phenotype in RPL.

In the present study, we evaluated the association of thrombophilia-related SNVs with coagulation parameters, with the specific aim of quantifying their relationship with key metabolic and hemostatic biomarkers. F7-rs6046 was significantly associated with alterations in INR or PT, and F12-rs1801020 is associated with differences in FXII activity and aPTT in the same cohort. By combining a multifactorial genotyping panel with biochemical and hemostatic measurements, we seek to clarify whether these two SNVs may influence hemostatic balance in the RPL setting, thus offering mechanistic and potentially translational insight into a challenging and incompletely understood cause of pregnancy loss.

## 2. Materials and Methods

### 2.1. Study Design and Patients

A total of 105 women were recruited between July 2021 and April 2024 as part of a retrospective descriptive study carried out in the National Institute of Perinatology in Mexico City. The origin noted in the patients was given by their birth status, but no genetic markers were used to denote ancestry. The inclusion criteria were diagnosis of RPL (defined as having experienced two or more consecutive spontaneous miscarriages before 20–24 weeks of gestation), genomic profile data for SNVs related to thrombophilia and coagulation parameters taken before the initiation of anticoagulant or antiplatelet therapy or any supplemental treatment reported by the patient.

Additionally, gynecological and obstetric history, along with potential contributing factors to RPL—including anatomical, endocrine, genetic, rheumatologic, hematologic, or idiopathic (if no other causes were identified)—were documented. The women without baseline coagulation parameters measurements or a previous clinical diagnosis of thrombophilia were excluded. All women initially included met the specifications and had at least six measurements in key metabolites, so none were eliminated. Since the genomic thrombophilia panel is an external study to the Institute, a socio-economic bias could be present.

A control group was not included because thrombophilia-related SNV testing is performed only in women with clinically confirmed RPL at our institution; thus, equivalent genomic and metabolite data are not available for individuals without RPL.

This protocol was approved by the Institutional Research Coordination of the Board of Education in Health Sciences on 18 September 2024 (registration number 2024.260). All participants had previously provided informed consent for the use of their data in academic and research contexts, in accordance with the General Law on Protection of Personal Data Held by Obligated Subjects, Article 22, Sections V and VII.

### 2.2. Metabolite Analysis: Coagulation and Hemostatic Biomarkers

Peripheral blood samples were collected after fasting. Metabolites were analyzed using commercially available reagents and protocols (Stago) and included fibrinogen (mg/dL), D-dimer (ng/mL), antithrombin III activity (%), thrombin time (TT, sec), prothrombin time (PT, sec), activated partial thromboplastin time (aPTT, sec), and the international normalized ratio (INR). Levels of folic acid (vitamin B9, ng/mL), cobalamin (vitamin B12, pg/mL), homocysteine (μmol/L), and factor XII activity (%) were assessed in an external laboratory when clinically indicated.

### 2.3. Genotyping

Genomic DNA was extracted from peripheral blood using standard protocols. SNVs were genotyped using real-time PCR with custom TaqMan probes. SNVs analyzed were angiotensinogen (AGT: rs4762, rs699), factor VII (F7: rs6046), fibrinogen (FGB: rs1800790), 5-methyl-tetrahydrofolate-homocysteine S-methyltransferase (MTR: rs1805087), methionine synthase reductase (MTRR: rs1801394), 5,10-methylenetetrahydrofolate reductase (MTHFR: rs1801133, rs1801131), factor II (F2: rs1799963), factor V (F5: rs6025), plasminogen activator inhibitor-1 (SERPINE1: rs1799889), factor XII (F12: rs1801020) and the A1 subunit of factor XIII (F13A1: rs5985). F12-rs1801020 and F13A1-rs5985 were not part of the panel design in 2021 and were included in 2022.

### 2.4. Statistical Methods

Quantitative variables’ distribution was assessed using the Shapiro–Wilk test. Allelic and genotypic frequencies were compared using Fisher’s exact test to assess for Hardy–Weinberg equilibrium to minimize bias in relation to sampling. Nonparametric tests (Mann–Whitney U or Kruskal–Wallis) were used to compare metabolite levels across genotypes. The *p*-value of <0.01 was considered statistically significant. Significant pairwise differences were further explored using linear regression under various genetic inheritance models (codominant, dominant, recessive, overdominant, additive). Pearson correlation coefficients were calculated to assess linear relationships. Analyses of haplotypes patterns were not performed because the theoretical number of multi-locus genotype combinations would exceed 1.5 million (3^13^), far surpassing the size of our cohort; therefore, statistical evaluation was restricted to single-locus associations. Statistical analyses were conducted using SPSS Statistics (version 29.0.2.0; IBM Corp.) and SNPStats were used for inheritance model testing [9]. Genotype distributions were visualized using Finetti plots generated with the HardyWeinberg package in R (via Datalab).

## 3. Results

The study population included 105 women with a mean age of 32.5 years (range, 19–42 years; SD, 3.9). Obstetric histories revealed a mean of 3.3 pregnancies (range, 2–7; SD, 1.18) and a mean of 2.9 miscarriages (range, 2–7; SD, 0.97). Nine women (8.6%) had a history of at least one vaginal delivery, thirteen (12.4%) had undergone cesarean section, two (1.9%) had a history of molar pregnancy, and twelve (11.4%) had experienced at least one ectopic pregnancy.

The mean number of contributing factors for RPL was 2.51 (range: 0–5, standard deviation: 0.901), with only three patients (2.85%) showing no factors associated with RPL. The most prevalent contributing factors were endocrine factors (82%) and anatomical factors (53.6%). The least frequent contributing factors included rheumatological (8.7%), chromosomal structural abnormalities (5.7%), and hematological ones (6%).

Biochemical analysis showed that mean metabolite values were generally within the reference ranges defined by the testing laboratories, except for folic acid (mean, 22.15 ng/mL; reference range, 3.1–20.5 ng/mL) and thrombin time (mean, 15.94 s reference range, 13–15.8 s), which exceeded the upper reference limit. Most metabolite values exhibited broader ranges than the reference intervals, though homocysteine levels remained within normal limits (range, 5.0–13.8 μmol/L; reference range, 5–15 μmol/L) (Appendix A).

### Genotype-Metabolite Associations

Allelic and genotype frequencies are shown in Appendix A. Ten SNVs were within Hardy–Weinberg equilibrium, the SNVs MTR-rs1805087 and SERPINE1-rs1799889, which showed heterozygote excess, and F2-rs1799963, which showed a heterozygote deficit did not meet Hardy–Weinberg equilibrium. These deviations are visualized in the Finetti diagram (Appendix A).

A statistically significant difference in INR was observed across genotypes of the F7-rs6046 SNV, being higher in the AA homozygous compared to the CC individuals, (*p =* 0.006; Figure 1A, Table 1), although the values remained within normal limits. Linear regression analysis using an additive model revealed a mean increase in INR of 0.05 with the number of alleles A (95% CI, 0.02–0.07; *p =* 0.0009) (Appendix A). Although a slight trend in PT was observed among the rs6046 genotypes (*p =* 0.179; Figure 1B), this effect did not reach statistical significance.

A significant association was observed between the F12-rs1801020 variants and factor XII activity being lower in the AA homozygous compared to the GA heterozygous, (*p =* 0.002; Figure 2A). Linear regression under an additive model revealed a mean difference of 24.92% in factor XII activity in the aforementioned genotypes (95% CI, 14.30–35.54%; *p =* 0.0001, Appendix A). aPTT, also linked to the intrinsic pathway, varied significantly according to genotype showing higher values in the group of homozygous individuals for the A allele compared to the homozygous for the ancestral G allele, (*p =* 0.045; Figure 2B), with an average difference of 2.17 s (95% CI, 0.50–3.84; *p =* 0.013) under an additive model. Despite these associations, Pearson’s correlation coefficient between Factor XII activity and aPTT did not reach statistical significance (r = −0.30, *p =* 0.107). The linear regression equation (y = −0.1x + 40.27) suggests that a 10% increase in Factor XII activity corresponds to a 1 s reduction in aPTT, with an R^2^ of 0.090 (Appendix A).

No statistically significant differences were observed in metabolite distributions across genotypes for the remaining eleven SNVs, so they were not analyzed by linear regression. Detailed results, including additional genotype-metabolite distribution comparisons, are provided in Table 1.

## 4. Discussion

The present study emphasizes the relationship between thrombophilia-related SNVs and their downstream on coagulation and biochemical parameters, in contrast to most previous studies that primarily focused on the association between these genetic variants and RPL [10,11]. All coagulation parameters analyzed in this study correspond to pre-treatment baseline values, as all participants were evaluated before starting any anticoagulant or antiplatelet therapy, thereby eliminating the potential confounding influence of heparin or related treatments. More than 80% of participants originated from the metropolitan region, which may limit the generalizability of our findings to the broader Mexican population. The genomic panel employed was designed under the hypothesis that thrombophilia-associated SNVs may exacerbate the prothrombotic state characteristic of pregnancy, promoting placental thrombosis and compromising maternal-fetal exchange [12]. Among the evaluated SNVs, two showed statistically significant associations with hemostatic markers: rs6046 in the F7 gene, part of the extrinsic coagulation pathway, and rs1801020 in F12, a key player in the intrinsic pathway.

The rs6046 variant in F7 has been functionally linked to reduced factor VII activity—up to 30% in heterozygotes and 43% in homozygotes [13]. F7 encodes a protein essential for the initiation of coagulation via tissue factor binding and subsequent activation of factor X [14]. The minor allele frequency (MAF) of rs6046 in our cohort (0.1285) closely matched that reported globally (0.1265) (7), although it differs from that observed in genetic variation observed in individuals from the Mexico City Prospective Study (MCPS). Soria et al. [15] demonstrated that genetic variability in F7 contributes significantly to factor VII activity, with seven variants accounting collectively for over one-third of its phenotypic variance, reinforcing its polygenic regulation. Although we observed a slight trend toward PT with increasing rs6046 risk alleles (*p =* 0.179), INR was significantly elevated although it remained within normal limits. This finding complements prior research showing that rs6046 may offer protective effects in RPL; one study reported an OR of 0.35 (95% CI, 0.23–0.53; *p* < 0.0001) [7]. These effects may arise from a relative attenuation of coagulation in a hypercoagulable milieu, rather than from an abnormal prolongation of INR, consistent with what we observed in our cohort.

In contrast, the rs1801020 variant in F12 demonstrated robust associations with both factor XII activity and aPTT, in line with previous reports. F12 encodes the zymogen FXII, which plays roles not only in intrinsic coagulation but also in fibrinolysis, inflammation, and complement activation [16,17,18]. Calafell et al. [19] identified rs1801020 as the principal determinant of FXII activity in a Spanish cohort. This SNV may affect translation through two proposed mechanisms: (1) creation of an upstream start codon resulting in a truncated peptide [19,20,21], or (2) disruption of the Kozak sequence impairing translation initiation [22,23,24,25].

Our findings confirmed a significant genotype-dependent reduction in FXII activity. In comparison with published data from Kanaji et al. [22] and Calafell et al. [19], our activity values for heterozygotes (C/T = 81.8%) and homozygotes (T/T = 56.9%) closely mirror those in thrombosis-related cohorts, despite not measuring wild-type individuals (C/C). The aPTT prolongation observed in our cohort (mean difference, 2.17 s; 95% CI, 0.50–3.84) under an additive model also exceeded that reported in the ARIC study by Weng et al. [26], where rs1801020 accounted for 11–12% of aPTT variance in both European and African Americans. The stronger effect observed here may be explained by phenotype-based cohort enrichment or population-specific genetic architecture, warranting further investigation.

Interethnic variability in rs1801020 allele frequency is well documented. The T allele is found at frequencies of 0.18 in Spanish, 0.25 in Dutch [24,27], and up to 0.73 in Japanese populations [22,24]. In our cohort, MAF reached 0.44, closely resembling that of African ancestry groups [26] and close to the genetic variation observed in individuals from the MCPS (0.48). While this may reflect a phenotype-driven enrichment, future studies comparing this frequency in population-based Mexican cohorts are needed to determine whether this variant contributes to RPL risk at the population level.

Although this is the first study in Mexico to evaluate associations between thrombophilia-related SNVs and their metabolic correlates in RPL, prior work by Domínguez et al. [18] assessed factor XII activity and aPTT in venous thromboembolism. Their reported values were similar to those in our cohort, although their broader measurement range, particularly for FXII activity, may reflect the inclusion of wild-type individuals.

Our findings suggest that rs1801020 contributes to variation in FXII activity and aPTT prolongation but is unlikely to explain the RPL phenotype. These data support the concept that the SNV under investigation may act as contributory—not singular—factors in RPL. The observed genotype–phenotype relationships should therefore be interpreted as contributory signals within the broader multifactorial landscape of RPL, rather than as direct causal determinants. Their pathogenic effects likely depend on broader genetic context, metabolic state, and pregnancy-specific physiological conditions. Although our cohort consisted exclusively of non-pregnant women, it is important to situate these findings within the broader physiologic context: pregnancy is a well-established hypercoagulable state in which several hemostatic components are upregulated while routine coagulation tests may still fall within normal ranges [28]. These emerging associations may pave the way for clinically relevant, personalized interventions tailored to the allelic profiles of women affected by RPL. Although such approaches could eventually include targeted therapeutic strategies—such as the selective use of low-molecular-weight heparin in specific genetic or biochemical contexts—additional prospective studies are required before any evidence-based clinical recommendations can be established.

Finally, the retrospective design, the possible socioeconomic bias, and the metropolitan recruitment without measurement of genetic markers of ancestry in our cohort limit the generalizability of these results. Prospective studies incorporating participants from diverse geographic regions across the country are needed to overcome current design limitations and enhance the generalizability and impact of the findings. Metabolomics research could also be expanded to involve other SNVs related to inflammatory or other related disease pathways to benefit personalized medicine in RPL.

## 5. Conclusions

This study represents the first effort in Mexico to investigate the relationship between thrombophilia-related SNVs and associated coagulation parameters in women with RPL. The primary outcome was the identification of significant associations between two SNVs, F7-rs6046 and F12-rs1801020, and alterations in coagulation parameters, specifically INR and aPTT, in Mexican women with RPL. These findings support the notion that specific genetic variants involved in coagulation pathways may modulate coagulation processes and contribute to a prothrombotic state that impairs placental function and, ultimately, pregnancy viability.

The observed effects were consistent with previous functional evidence and appear more pronounced in this phenotypically enriched cohort, underscoring the potential value of integrating genetic and biochemical data in the clinical evaluation of RPL. In this context, the results suggest that integrating genetic screening into the clinical evaluation of RPL may enhance personalized risk stratification and inform targeted therapeutic strategies.

Overall, this study contributes to foundational knowledge on the pathophysiology of RPL and highlights the importance of a comprehensive and systematic approach to its etiology—particularly in genetically diverse populations. Given the multifactorial nature of thrombophilia and its variable expression across individuals, future research in broader, unselected cohorts is warranted to validate these findings and further explore their clinical applicability.

## Figures and Tables

**Figure 1 diagnostics-15-03111-f001:**
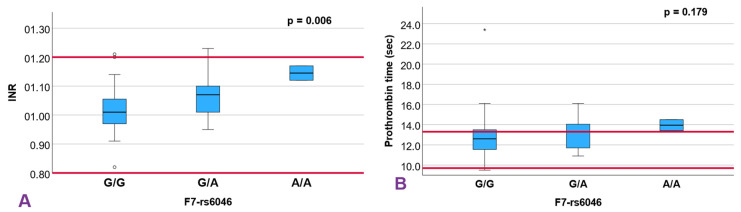
Box plot showing the distribution of INR values (**A**) and prothrombin time values (**B**) among the SNV *F7*-rs6046 genotypes analyzed using the Kruskal–Wallis test for independent samples. The horizontal lines of references values are highlighted in red. (*) Refers to extremely atypical values. (León, 2024, IBM SPSS Statistics).

**Figure 2 diagnostics-15-03111-f002:**
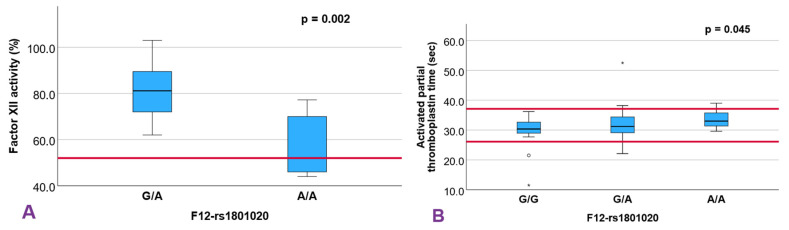
Box plot showing the distribution of factor XII activity (**A**) and activated partial thromboplastin time values (**B**) among the SNV *F12*-rs1801020 genotypes analyzed using the Kruskal–Wallis test for independent samples. The horizontal lines of references values are highlighted in red. (*) Refers to extremely atypical values. (León, 2024, IBM SPSS Statistics).

**Table 1 diagnostics-15-03111-t001:** Results of the analysis comparing the distribution of metabolite levels across different genotypes of the analyzed SNV.

	Metabolite	B9	B12	H	F	AIII	TT	TP	TTPA	DD	INR	FXII
SNV												
*AGT* rs4762		0.151	0.731	0.208	0.47	0.602	0.521	0.993	0.594	0.712	0.705	0.846
*AGT* rs699		0.044 *	0.547	0.827	0.200	0.475	0.196	0.631	0.745	0.042 *	0.423	0.139
*F7* rs6046		0.233	0.251	0.498	0.628	0.266	0.865	0.179	0.930	0.760	0.006 **	0.105
*FGB* rs1800790		0.266	0.863	0.586	0.453	0.946	0.361	0.157	0.335	0.296	0.166	0.667
*MTR* rs1805087		0.783	0.351	0.107	0.111	0.840	0.453	0.550	0.871	0.956	0.641	0.907
*MTRR* rs1801394		0.521	0.466	0.326	0.913	0.331	0.436	0.998	0.027 *	0.271	0.946	0.929
*MTHFR* rs1801133		0.076	0.337	0.390	0.023 *	0.572	0.805	0.738	0.926	0.675	0.908	0.192
*MTHFR* rs1801131		0.314	0.698	0.513	0.295	0.511	0.049 *	0.032 *	0.410	0.410	0.066	0.392
*F2* ^1^ rs1799963		0.344	0.031 *	1.000	0.788	0.038 *	0.735	0.076	0.171	0.667	0.824	NC
*F5* ^1^ rs6025		0.906	0.123	0.895	0.611	0.525	0.342	0.449	0.844	0.776	1.000	0.133
*F12* rs1801020		0.651	0.638	0.404	0.204	0.315	0.223	0.506	0.045 *	0.397	0.416	0.002 **
*F13A1* rs5985		0.425	0.700	0.954	0.409	0.442	0.730	0.833	0.587	0.245	0.841	0.376
*SERPINE1* rs1799889		0.343	0.442	0.142	0.244	0.140	0.363	0.486	0.368	0.548	0.850	0.785

NOTE: All SNV were analyzed using the Kruskal–Wallis test for independent samples except those marked (1), which were analyzed using the Mann–Whitney U test. B9: Folic acid, B12: Cobalamin, H: Homocysteine, F: Fibrinogen, AIII: Antithrombin III, TT: Thrombin time, PT: Prothrombin time, APTT: Activated partial thromboplastin time, DD: D-dimer, FXII: Factor XII activity, NC: Calculation not possible because one of the groups had missing data. (*): Statistical significance at *p* < 0.05. (**): Statistical significance at *p* < 0.01

## Data Availability

The original contributions presented in this study are included in the article/Appendix A. Further inquiries can be directed to the corresponding authors.

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
