# Peer review of "Thrombophilia-Related Single Nucleotide Variants and Altered Coagulation Parameters in a Cohort of Mexican Women with Recurrent Pregnancy Loss"

_diagnostics, 2025, doi:10.3390/diagnostics15243111_

Round 1

Reviewer 1 Report

Comments and Suggestions for Authors

In this work authors correlated thrombophilia-related single nucleotide variants (SNVs) and coagulation-related metabolites in a 105 Mexican women with recurrent pregnancy loss.

-in Materials and methods you wrote:”A total of 105 women were recruited between July 2021 and April 2024 as part of a retrospective descriptive study. The inclusion criteria were diagnosis of RPL, genomic profile data for SNVs related to thrombophilia and coagulation parameters taken before the initiation of anticoagulant or antiplatelet therapy or any supplemental treatment reported by the patient were included. The women without baseline coagulation parameters measurements or a previous clinical diagnosis of thrombophilia were excluded.”Ok. This is the study group. What about the control group? You have no control group. Please explain why.  

-in Abstract you wrote:”Methods: A retrospective and descriptive design was conducted including 105 women with at least two consecutive miscarriages.”Ok. Please add “and previously diagnosed with thrombophilia”

-you wrote:”From this perspective, two single-nucleotide variants assume relevance, on gene F7 the SNV rs6046 (Arg353Gln, also known as R353Q) this missense variant on the coagulation factor VII has been extensively studied in cardiovascular and thrombosis research.” Please cut into three sentences and rephrase. This is only one. The second one is in the last sentences :”In addition, the SNV rs1801020 in the gene F12, has been mechanistically linked to reduced translation and lower Factor XII levels/activity, and genome-wide association studies have shown that rs1801020 contributes significantly to interindividual variation in aPTT and FXII activity [1]. Yet, its direct role in RPL remains underexplored. To the best of our knowledge, there are no studies testing the association of the SNV rs1801020 with RPL.”Please rephrase the first sentence. It is not clear.

-In page 3 please remove:”.The introduction should briefly place the study in a broad context and highlight why it is important. It should define the purpose of the work and its significance.The current state of the research field should be carefully reviewed and key publications cited. Please highlight controversial and diverging hypotheses when necessary. Finally, briefly mention the main aim of the work and highlight the principal conclusions. As far as possible, please keep the introduction comprehensible to scientists outside your particular field of research. References should be numbered in order of appearance and indicated by a numeral or numerals in square brackets—e.g., [1] or [2,3], or [4–6]. See the end of the document for further details on references”

-in Materials and Methods, in “The inclusion criteria were diagnosis of RPL, genomic profile data for SNVs related to thrombophilia and coagulation parameters taken before the initiation of anticoagulant or antiplatelet therapy or any supplemental treatment reported by the patient were included”, please remove the last two words:”were included”.

-in Results you wrote:”Biochemical analysis showed that mean metabolite values were generally within the reference ranges defined by the testing laboratories, except for folic acid (mean, 22.15 ng/mL; reference range, 3.1–20.5 ng/mL) and thrombin time (mean, 15.94 sec reference range, 13–15.8 sec), which exceeded the upper reference limit”Ok. Within normal limits all of them. Ok. Later in Results you wrote:”A statistically significant difference of INR was observed across genotypes of the F7-  rs6046 SNV, being higher in the AA homozygous compared to the CC individuals, (p =0.006; Figure 1A, Table 1).” It may be statistically significant,but INR is still within normal limits, as seen in Figure 1, too. Please write that INR is still within the normal limits.

-then you wrote:”A significant association was observed between the F12-rs1801020 variants and factor XII activity being lower in the AA homozygous compared to the GA heterozygous, (p =0.002; Figure 2A).” Ok.  But still in the normal limits, as seen in Figure 2A, too. Please write that, too.

-then you wrote:”Linear regression under an additive model revealed a mean difference of 24.92% in factor XII activity in the aforementioned genotypes (95% CI, 14.30–35.54%; p= 0.0001, Supplementary Table 4). aPTT, also linked to the intrinsic pathway, varied significantly according to genotype showing higher values in the group of homozygous individuals for the A allele compared to the homozygous for the ancestral G allele, (p = 0.045; Figure 2B), with an average difference of 2.17 seconds (95% CI, 0.50–3.84; p = 0.013) under an additive model.” True. But still within normal limits. As seen in Figure 2B.Please write that, too.

-in Discussion you wrote:”Although we observed a slight trend toward PT with increasing rs6046 risk alleles (p = 0.179), INR was significantly elevated.”. True. But INR still within normal limits.Please write that.

-then you wrote:”This finding complements prior research showing that rs6046 may offer protective effects in RPL; one study reported an OR of 0.35 (95% CI, 0.23–0.53; p < 0.0001) [16]. These effects may arise from modest downregulation of coagulation in a hypercoagulable environment, consistent with INR prolongation in our cohort.”What protection? INR is still within normal limits.

-you wrote:”The aPTT prolongation observed in our cohort (mean difference, 2.17 seconds; 95% 287 CI, 0.50–3.84) under an additive model also exceeded that reported in the ARIC study by Weng et al. [27], where rs1801020 accounted for 11%–12% of aPTT variance in both European and African Americans. The stronger effect observed here may be explained by phenotype-based cohort enrichment or population-specific genetic architecture, warranting further investigation.”Actually, treatment may be involved here:

-all the patients you studied already had a diagnosis of thrombophilia. And pregnancy. Most of them, I assume, were under treatment for thrombophilia, probably heparin, which INFLUENCED AND NORMALIZED coagulation parameters. Since you said nothing about heparin or something else as treatment already taken by your patients, you should say AT LEAST ONE PARAGRAPH in Discussion, showing that HEPARIN TREATMENT did influence coagulation factors a lot. So that one HUGE weak point is that you did not associate the SNVs and coagulation factors with the heparin treatment.

-in Introduction or in Discussion you could start saying something about the hypercoagulable state that already exists in all pregnant patients, still coagulation factors look normal in many of them. You could cite some other articles about this, like:

-Covali R, Socolov D, Socolov R. Coagulation tests and blood glucose before vaginal delivery in healthy teenage pregnant women compared with healthy adult pregnant women. Medicine (Baltimore). 2019 Feb;98(5):e14360. doi: 10.1097/MD.0000000000014360. PMID: 30702627; PMCID: PMC6380794.

- please remove “Informed Consent Statement: “Informed consent was obtained from all subjects involved in the study.”, since you wrote it twice.

-there are 28 titles in References. Only 12 are recent. They are well chosen. No inappropriate self-citation detected.

Author Response

Thank you very much for taking the time to review this manuscript. Please find the detailed responses below.

Point-by-point response to Comments and Suggestions for Authors

Comments 1: -in Materials and methods you wrote:”A total of 105 women were recruited between July 2021 and April 2024 as part of a retrospective descriptive study. The inclusion criteria were diagnosis of RPL, genomic profile data for SNVs related to thrombophilia and coagulation parameters taken before the initiation of anticoagulant or antiplatelet therapy or any supplemental treatment reported by the patient were included. The women without baseline coagulation parameters measurements or a previous clinical diagnosis of thrombophilia were excluded.”Ok. This is the study group. What about the control group? You have no control group. Please explain why.

Response 1: Thank you for this observation. As now stated in the revised Methods section, a control group was not included because thrombophilia-related SNV testing is performed exclusively in women with clinically confirmed RPL at our institution, and therefore comparable genomic and metabolite data from women without RPL are not available. This clarification, explaining the absence of a control group, has been explicitly added to the Methods section. The aim of the present study was not to compare SNV frequencies between women with and without RPL, but rather to examine whether, within this clinically defined cohort, different genotypes were associated with variations in coagulation-related metabolites. All SNVs analyzed conformed to Hardy–Weinberg equilibrium, supporting an expected distribution within the sample and allowing meaningful comparison of metabolite values across the three genotype groups (wild-type homozygotes, heterozygotes, and alternate homozygotes).

This design aligns with the objectives of the study, which focuses on genotype–metabolite associations rather than on case–control analyses of SNV prevalence.

Comments 2: in Abstract you wrote:”Methods: A retrospective and descriptive design was conducted including 105 women with at least two consecutive miscarriages.”Ok. Please add “and previously diagnosed with thrombophilia”

Response 2: We appreciate the reviewer’s suggestion. The Abstract has been revised accordingly, and the phrase has been added as requested.

Comments 3: -you wrote:”From this perspective, two single-nucleotide variants assume relevance, on gene F7 the SNV rs6046 (Arg353Gln, also known as R353Q) this missense variant on the coagulation factor VII has been extensively studied in cardiovascular and thrombosis research.” Please cut into three sentences and rephrase. This is only one. The second one is in the last sentences :”In addition, the SNV rs1801020 in the gene F12, has been mechanistically linked to reduced translation and lower Factor XII levels/activity, and genome-wide association studies have shown that rs1801020 contributes significantly to interindividual variation in aPTT and FXII activity [1]. Yet, its direct role in RPL remains underexplored. To the best of our knowledge, there are no studies testing the association of the SNV rs1801020 with RPL.”Please rephrase the first sentence. It is not clear.

Response 3: We thank the reviewer for this helpful observation. The paragraph has been revised as requested: the long sentence describing the F7 rs6046 variant was divided and rephrased for clarity, and the sentence concerning F12 rs1801020 has been rewritten to ensure clearer expression of its mechanistic relevance. These changes have been incorporated into the updated version of the manuscript.

Comments 4: -In page 3 please remove:”.The introduction should briefly place the study in a broad context and highlight why it is important. It should define the purpose of the work and its significance.The current state of the research field should be carefully reviewed and key publications cited. Please highlight controversial and diverging hypotheses when necessary. Finally, briefly mention the main aim of the work and highlight the principal conclusions. As far as possible, please keep the introduction comprehensible to scientists outside your particular field of research. References should be numbered in order of appearance and indicated by a numeral or numerals in square brackets—e.g., [1] or [2,3], or [4–6]. See the end of the document for further details on references”

Response 4: Thank you for pointing this out. The template text on page 3 has been removed entirely, as requested.

Comments 5: -in Materials and Methods, in “The inclusion criteria were diagnosis of RPL, genomic profile data for SNVs related to thrombophilia and coagulation parameters taken before the initiation of anticoagulant or antiplatelet therapy or any supplemental treatment reported by the patient were included”, please remove the last two words:”were included”.

Response 5: We appreciate the reviewer’s observation. The phrase “were included” has been removed from the corresponding sentence in the Materials and Methods section as requested.

Comments 6: -in Results you wrote:”Biochemical analysis showed that mean metabolite values were generally within the reference ranges defined by the testing laboratories, except for folic acid (mean, 22.15 ng/mL; reference range, 3.1–20.5 ng/mL) and thrombin time (mean, 15.94 sec reference range, 13–15.8 sec), which exceeded the upper reference limit”Ok. Within normal limits all of them. Ok. Later in Results you wrote:”A statistically significant difference of INR was observed across genotypes of the F7- rs6046 SNV, being higher in the AA homozygous compared to the CC individuals, (p =0.006; Figure 1A, Table 1).” It may be statistically significant,but INR is still within normal limits, as seen in Figure 1, too. Please write that INR is still within the normal limits.

Response 6: Thank you for this helpful clarification. The Results section has been updated to explicitly state that, although the observed differences in INR across F7-rs6046 genotypes were statistically significant, all INR values remained within the established normal reference range.

Comments 7: -then you wrote:”A significant association was observed between the F12-rs1801020 variants and factor XII activity being lower in the AA homozygous compared to the GA heterozygous, (p =0.002; Figure 2A).” Ok. But still in the normal limits, as seen in Figure 2A, too. Please write that, too.

Response 7: We thank the reviewer for this observation. We carefully reviewed the reference range and the distribution of Factor XII activity values across F12-rs1801020 genotypes. Unlike the INR results, the values for FXII activity in homozygous individuals fall below the lower reference limit (52%), as can be appreciated in Figure 2A. Therefore, in this case we could not state that FXII activity remained fully within normal limits, and this distinction has been maintained in the revised manuscript to accurately reflect the data.

Comments 8: -then you wrote:”Linear regression under an additive model revealed a mean difference of 24.92% in factor XII activity in the aforementioned genotypes (95% CI, 14.30–35.54%; p= 0.0001, Supplementary Table 4). aPTT, also linked to the intrinsic pathway, varied significantly according to genotype showing higher values in the group of homozygous individuals for the A allele compared to the homozygous for the ancestral G allele, (p = 0.045; Figure 2B), with an average difference of 2.17 seconds (95% CI, 0.50–3.84; p = 0.013) under an additive model.” True. But still within normal limits. As seen in Figure 2B. Please write that, too.

Response 8: We appreciate the reviewer’s comment. We reviewed the reference intervals and the distribution of aPTT values across F12-rs1801020 genotypes. While the mean values remain near the upper limit of normal, the distribution shown in Figure 2B demonstrates that, for both the heterozygous and homozygous alternate genotypes, the upper quartile exceeds the reference range. Therefore, it would not be accurate to state that all aPTT values remained within normal limits. This distinction has been preserved in the revised manuscript to faithfully reflect the observed data.

Comments 9: -in Discussion you wrote:”Although we observed a slight trend toward PT with increasing rs6046 risk alleles (p = 0.179), INR was significantly elevated.”. True. But INR still within normal limits. Please write that.

Response 9: Thank you for the clarification. The Discussion section has been updated to explicitly state that, although INR values were significantly higher across rs6046 genotypes, they nonetheless remained within the normal reference range.

Comments 10: -then you wrote:”This finding complements prior research showing that rs6046 may offer protective effects in RPL; one study reported an OR of 0.35 (95% CI, 0.23–0.53; p < 0.0001) [16]. These effects may arise from modest downregulation of coagulation in a hypercoagulable environment, consistent with INR prolongation in our cohort.”What protection? INR is still within normal limits.

Response 10: We thank the reviewer for this insightful comment. We agree that INR values in our cohort remained within normal reference limits. However, it is important to note that all participants were non-pregnant at the time of sampling, and pregnancy is a physiologically hypercoagulable state. Thus, even a modest INR prolongation—while still normal—may reflect a relative downregulation of coagulation potential. This interpretation aligns with previous reports suggesting that rs6046 may exert a protective effect in the context of RPL. The text has been revised to clarify that the potential protective influence refers to a relative attenuation of coagulation in a hypercoagulable milieu, rather than an abnormal prolongation of INR.

Comments 11: -you wrote:”The aPTT prolongation observed in our cohort (mean difference, 2.17 seconds; 95% 287 CI, 0.50–3.84) under an additive model also exceeded that reported in the ARIC study by Weng et al. [27], where rs1801020 accounted for 11%–12% of aPTT variance in both European and African Americans. The stronger effect observed here may be explained by phenotype-based cohort enrichment or population-specific genetic architecture, warranting further investigation.”Actually, treatment may be involved here:

-all the patients you studied already had a diagnosis of thrombophilia. And pregnancy. Most of them, I assume, were under treatment for thrombophilia, probably heparin, which INFLUENCED AND NORMALIZED coagulation parameters. Since you said nothing about heparin or something else as treatment already taken by your patients, you should say AT LEAST ONE PARAGRAPH in Discussion, showing that HEPARIN TREATMENT did influence coagulation factors a lot. So that one HUGE weak point is that you did not associate the SNVs and coagulation factors with the heparin treatment.

Response 11: We appreciate the reviewer’s concern regarding the potential influence of anticoagulant therapy—particularly heparin—on coagulation parameters. We would like to clarify that none of the patients included in this study were pregnant at the time of sampling, and, importantly, all biochemical and hematologic measurements were obtained prior to the initiation of any anticoagulant or antiplatelet therapy, including heparin, as detailed in the Materials and Methods section. This criterion was explicitly incorporated to avoid confounding effects of treatment on coagulation parameters. Therefore, the observed associations between SNVs and hemostatic biomarkers reflect untreated baseline physiology, independent of pharmacologic modulation. We have added a brief statement in the Discussion to emphasize that all measurements reflect pre-treatment values, ensuring that anticoagulant therapy did not influence the reported results.

Comments 12: -in Introduction or in Discussion you could start saying something about the hypercoagulable state that already exists in all pregnant patients, still coagulation factors look normal in many of them. You could cite some other articles about this, like: -Covali R, Socolov D, Socolov R. Coagulation tests and blood glucose before vaginal delivery in healthy teenage pregnant women compared with healthy adult pregnant women. Medicine (Baltimore). 2019 Feb;98(5):e14360. doi: 10.1097/MD.0000000000014360. PMID: 30702627; PMCID: PMC6380794.

Response 12: We appreciate the reviewer’s suggestion to contextualize our findings within the known hypercoagulable state of pregnancy. Although our study included only non-pregnant women, we agree that this physiologic framework is relevant. Accordingly, we have added a brief statement addressing this point, and the reference provided by the reviewer has been incorporated at the end of the Discussion section.

Comments 13: - please remove “Informed Consent Statement: “Informed consent was obtained from all subjects involved in the study.”, since you wrote it twice.

Response 13: Thank you for pointing this out. The duplicated sentence regarding informed consent has been removed from the Institutional Review Board Statement, and it is now presented only once as a separate Informed Consent Statement, as required by the Journal.

Comments 14: -there are 28 titles in References. Only 12 are recent. They are well chosen. No inappropriate self-citation detected.

Response 14: We thank the reviewer for this positive assessment of our reference list. We are pleased to note that the selected citations were considered appropriate and sufficiently recent, and we appreciate the acknowledgement that no inappropriate self-citation was detected. The complete reviewed version is attached below. 

Reviewer 2 Report

Comments and Suggestions for Authors

In their manuscript "Thrombophilia-Related Single Nucleotide Variants and Altered Coagulation Parameters in a Cohort of Mexican Women with Recurrent Pregnancy Loss," Luis Felipe León-Madero et al. describe a comprehensive study investigating a possible correlation between two newly detected SNVs and recurrent pregnancy loss.

The study is retrospective and was conducted on 105 women in the Mexico City metropolitan area. This is a good approach and a lot of work, which my review does not wish to detract from. However, there is no mention whatsoever of whether these are 105 individuals who stood out in a complete analysis of the available data sets. Were individuals excluded? Are there exclusion criteria? Is there a bias that could have affected the selection process? Which hospitals were involved? Which institutes were involved? Is the study monocentric? Is there any evidence of socio-economic bias due to the selected collective? Is there any evidence of genetic origin/race?

The study is simple in design, with data collected retrospectively. In this respect, it may not be possible to glean any further information from the available data. However, there is a complete lack of information on comorbidities and other possible causes that could make RPL likely. Are there any indications of antiphospholipid syndrome? Was this ruled out in each case?

How common are SNV F7 and F12 in the general population that does not have RPL? Is there any data on this?

It makes sense to perform a Kruskal-Wallis test to demonstrate the independence of the data. However, I would like to see a clear presentation of the number of variants (SNVs) detected in the cohort studied. How often was a co-mutation present?

Can the supposed increase in the prevalence of SNV also be confirmed outside the metropolitan area? Is SNV prevalent throughout the entire territory of Mexico? There is no control group or even a control approach here. I welcome the opportunity for discussion and improvement. Otherwise, in my view, the lack of such consideration represents an unacceptable quality deficit.

Figure 1 and Figure 2 suggest high statistical validity and significance. I am missing the absolute numbers. How many individuals with SNV are reported here? Is there further evidence that causality can be found here? It may be that the detected SNV is responsible for a slight increase in INR, but absolute figures are needed to substantiate this claim.

As a very minor shortcoming, I would like to point out that, for layout reasons, the asterisk indicating the significance rating is always displayed on the same side of the box plot.

Under the circumstances, despite the extensive preparatory work, I unfortunately do not consider the manuscript sufficient for publication. The points of criticism should at least be discussed in detail.

Author Response

Thank you very much for taking the time to review this manuscript. Please find the detailed

responses below.

Point-by-point response to Comments and Suggestions for Authors

Comments 1: The study is retrospective and was conducted on 105 women in the Mexico City metropolitan area. This is a good approach and a lot of work, which my review does not wish to detract from. However, there is no mention whatsoever of whether these are 105 individuals who stood out in a complete analysis of the available data sets. Were individuals excluded? Are there exclusion criteria? Is there a bias that could have affected the selection process? Which hospitals were involved? Which institutes were involved? Is the study monocentric? Is there any evidence of socio-economic bias due to the selected collective? Is there any evidence of genetic origin/race?

Response 1: We thank the reviewer for these important observations regarding the composition and selection of the study cohort. In response, we have substantially expanded and clarified the Methods section. We now explicitly describe that the study included 105 women recruited between July 2021 and April 2024 at the National Institute of Perinatology in Mexico City, thus confirming that the study was monocentric. The inclusion criteria and clinical characteristics of the enrolled patients have been detailed, including the definition of RPL, availability of thrombophilia-related SNV genotyping, and baseline coagulation-metabolite measurements taken prior to any anticoagulant, antiplatelet, or supplemental therapy. We have also clarified that patients lacking these baseline measurements or with a prior clinical diagnosis of thrombophilia were excluded. Importantly, all women who initially met criteria had complete datasets so no participants were removed. To address the reviewer’s questions regarding ancestry and possible selection bias, we specify that while patients’ place of birth was recorded, no genetic markers of ancestry were used, and this is now acknowledged as a limitation. Additionally, because the thrombophilia genomic panel is payed by the patient, we note that some degree of socioeconomic bias may be present. To further clarify how sampling-related bias was evaluated, we have added that Hardy–Weinberg equilibrium was assessed using Fisher’s exact test as part of the statistical analysis. These clarifications have been incorporated into the revised manuscript to improve transparency and methodological rigor.

Comments 2: The study is simple in design, with data collected retrospectively. In this respect, it may not be possible to glean any further information from the available data. However, there is a complete lack of information on comorbidities and other possible causes that could make RPL likely. Are there any indications of antiphospholipid syndrome? Was this ruled out in each case?

Response 2: We appreciate the reviewer’s insightful comment regarding comorbidities and alternative causes of RPL. In the revised manuscript, we have clarified that all patients underwent a comprehensive clinical evaluation that included anatomical, endocrine, genetic, rheumatologic (including antiphospholipid syndrome), hematologic, and idiopathic categories, as now described in the Methods section. Contributing factors identified in each patient were also quantified and reported in the Results section, where we state the distribution and prevalence of these factors within the cohort. Regarding antiphospholipid syndrome (APS), we would like to clarify that APS testing is routinely performed as part of the rheumatologic assessment in our institutional RPL workup. Patients with confirmed APS were not excluded from the study; instead, APS was categorized within the rheumatologic contributing factors. This has now been made explicit in the manuscript. Importantly, the aim of the present study was not to establish SNVs as causal factors for RPL, but rather to evaluate whether thrombophilia-related SNVs are associated with alterations in key coagulation-related metabolites. This metabolic characterization represents an initial step toward identifying potential biological substrates that could inform future case–control studies designed to evaluate the relationship between these SNVs and RPL. These clarifications have been integrated to ensure full transparency regarding comorbidities and clinical background within the cohort.

Comments 3: How common are SNV F7 and F12 in the general population that does not have RPL? Is there any data on this?

Response 3: We thank the reviewer for raising this important question regarding the population frequency of the F7 and F12 SNVs. In response, we have incorporated the corresponding allele frequency data into the Discussion section. For F7-rs6046, we now compare the minor allele frequency observed in our cohort with global estimates and with available data from the Mexico City Prospective Study. For F12-rs1801020, we expanded the discussion to include interethnic variation and to contextualize our cohort’s allele frequency relative to published data and population-based information from Mexico. We would like to clarify that, while these population comparisons are informative, the primary aim of the present study was not to assess whether these SNVs are more frequent in women with RPL, but rather to determine whether they are associated with alterations in coagulation-related metabolites. Establishing comparative allele frequencies and evaluating their relevance to RPL risk will require future case–control studies specifically designed for that purpose, as now stated in the revised manuscript.

Comments 4: It makes sense to perform a Kruskal-Wallis test to demonstrate the independence of the data. However, I would like to see a clear presentation of the number of variants (SNVs) detected in the cohort studied. How often was a co-mutation present?

Response 4: We appreciate the reviewer’s interest in the distribution of variants within the cohort. Allelic and genotypic frequencies for all 13 SNVs analyzed are provided in Supplementary Table 2, which presents the Hardy–Weinberg equilibrium assessment for each locus. This table effectively documents the number of variants detected and their distribution across the sample. Regarding co-mutations, we did not perform haplotype or multi-locus combination analyses. With 13 SNVs and three possible genotypes per locus, the theoretical number of possible genotype combinations would be 3¹³ (>1.5 million), far exceeding the size of our cohort (n = 105), which would preclude meaningful or interpretable statistical inference. For this reason, and in alignment with the descriptive and metabolite-focused design of the study, we limited our analysis to single-locus effects. We have added a clarifying statement in the revised manuscript to explain this decision and to avoid any misunderstanding regarding multi-locus variation.

Comments 5: Can the supposed increase in the prevalence of SNV also be confirmed outside the metropolitan area? Is SNV prevalent throughout the entire territory of Mexico? There is no control group or even a control approach here. I welcome the opportunity for discussion and improvement. Otherwise, in my view, the lack of such consideration represents an unacceptable quality deficit.

Response 5: We thank the reviewer for this thoughtful and important comment. We agree that determining whether the prevalence of these SNVs is consistent across different regions of Mexico is a relevant question for future research. However, we would like to clarify that the primary aim of the present study was not to assess the population frequency or geographic distribution of these variants, but rather to evaluate whether thrombophilia-related SNVs are associated with alterations in coagulation-related metabolites within a clinically defined cohort of women with RPL. Furthermore, the thrombophilia panel, being a study that is not carried out unless the RPL condition is present, and due to the retrospective nature of the design, prevented comparisons from being made in a control group. For this reason, the study was not designed to include a population control group. We acknowledge this limitation and have addressed it in the revised Discussion. Nevertheless, because all SNVs analyzed conformed to Hardy–Weinberg equilibrium (Supplementary Table 2), their distribution within our cohort is consistent with expected allele frequencies for samples not affected by strong selection or sampling distortion. This equilibrium supported the validity of comparing metabolite levels across the three observed genotypes (wild-type homozygous, heterozygous, and alternate homozygous), which was the main analytic objective of the study. As now stated in the manuscript, establishing whether the prevalence of these SNVs differs across regions of Mexico—or whether they are enriched in women with RPL—will require future population-based or case–control studies specifically designed to evaluate allele frequency differences at a national level. Our findings instead provide an initial biological framework to support the design of such studies by identifying metabolite-level alterations potentially linked to these variants.

Comments 6: Figure 1 and Figure 2 suggest high statistical validity and significance. I am missing the absolute numbers. How many individuals with SNV are reported here? Is there further evidence that causality can be found here? It may be that the detected SNV is responsible for a slight increase in INR, but absolute figures are needed to substantiate this claim.

Response 6: We thank the reviewer for this important comment. While Figures 1 and 2 illustrate the distribution of metabolite values across genotypes, the absolute numbers and quantitative effect sizes associated with each SNV are provided in the supplementary material. Specifically, Supplementary Table 3 (for F7-rs6046 and INR) and Supplementary Table 4 (for F12-rs1801020 and FXII activity) present the sample size for each genotype group as well as the linear regression results, including mean differences, confidence intervals, and p-values. Regarding causality, we fully agree with the reviewer that the present study cannot—and does not attempt to—establish causal relationships between these SNVs and the measured coagulation parameters. As highlighted in the final section of the Discussion, these variants should be considered contributory rather than singular determinants within the multifactorial context of RPL. Our findings demonstrate statistically significant associations but do not support causal inference, and we have emphasized this point clearly in the manuscript.

Comments 7: As a very minor shortcoming, I would like to point out that, for layout reasons, the asterisk indicating the significance rating is always displayed on the same side of the box plot.

Response 7: We thank the reviewer for noting this minor layout issue. The placement of the significance asterisks in Figures 1 and 2 has been adjusted to improve their alignment with the corresponding boxplots and ensure clearer visual interpretation. The updated figures have been incorporated into the revised manuscript. The complete reviewed version is attached below. 

Reviewer 3 Report

Comments and Suggestions for Authors

In this study the authors aimed to investigate the association between thrombophilia-related single nucleotide variants (SNVs) and coagulation-related metabolites in a cohort of Mexican women with RPL. They observed that F7-rs6046 and F12-rs1801020 genotypes influence specific coagulation parameters, suggesting that certain thrombophilia-associated SNVs may modulate the hemostatic profile.

The manuscrip is clear and well written. I would suggest to revise the manuscript according to the following comments.

  1. Introduction: the aim of the study shouldbe clearly state in the text; the final part of the text should be removed as it is part of the template
  2. Methods: please clarify the definition of RPL and improve the description of the inclusion criteria (clinical characteristics of the included patients)
  3. Results: please organize the section in different paragraphs; 3.3 should be removed or  revised according to the other paragraphs; please integrate tables and figure in the main text
  4. Discussion: please provide potential implication of the findings in the diagnostic workflow of the patients undergoing Assisted Reproduction with a history of RPL

Author Response

Point-by-point response to Comments and Suggestions for Authors

Comments 1: 1. Introduction: the aim of the study shouldbe clearly state in the text; the final part of the text should be removed as it is part of the template.

Response 1: We appreciate the reviewer’s observation. In accordance with the recommendation, we have revised the final paragraph of the Introduction to clearly state the aim of the study. Additionally, the section that belonged to the template has been removed as suggested. This change can be found on line.

Comments 2: 2. Methods: please clarify the definition of RPL and improve the description of the inclusion criteria (clinical characteristics of the included patients).

Response 2: We thank the reviewer for this helpful suggestion. Although the definition of RPL was already provided in the Introduction, we have repeated it in the Methods section to ensure clarity. We have also expanded the description of the inclusion criteria by adding the relevant clinical characteristics studied of the patients included in the study. This change can be found on line.

Comments 3: 3. Results: please organize the section in different paragraphs; 3.3 should be removed or revised according to the other paragraphs; please integrate tables and figure in the main text

Response 3: We appreciate the reviewer’s insightful comment. The Results section has been reorganized into clearer, well-defined paragraphs, ensuring a more structured presentation of the findings. Additionally, we have revised the section to better integrate the figures and tables within the main text, providing explicit descriptions and alignment with the reported results.

Comments 4: 1. Discussion: please provide potential implication of the findings in the diagnostic workflow of the patients undergoing Assisted Reproduction with a history of RPL

Response 4: We thank the reviewer for this important observation. In response, we have expanded the Discussion to address the potential implications of our findings within the diagnostic workflow of patients undergoing Assisted Reproduction with a history of RPL. Specifically, we incorporated a statement highlighting how these genetic–metabolic associations may inform personalized clinical assessment and could ultimately guide therapeutic considerations—such as the selective use of low-molecular-weight heparin—while clearly noting that additional prospective studies are required before such interventions can be recommended. This addition aims to situate our results within a broader clinical context without overextending the evidence. This change can be found on line. The complete reviewed version is attached below. 

Round 2

Reviewer 2 Report

Comments and Suggestions for Authors

The authors have endeavored to provide a comprehensive discussion of the points of criticism mentioned. Many passages have also been supplemented, changed, or otherwise adapted. The socio-economic bias has also been discussed.
I recommend the publication of the manuscript in its current form.